# Significantly higher atherosclerosis risks in patients with obstructive sleep apnea and non-alcoholic fatty liver disease

**Samshol Sukahri**[1], **Fatimah Zaherah Mohamed Shah**[1], **Ahmad Izuanuddin Ismail**[1], **Marymol Koshy**[2], **Bushra Johari**[2], **Mazuin Mohd Razali**[2], **Thuhairah Hasrah Abdul Rahman**[3], **Mohamad Rodi Isa**[4], **Rohana Abdul Ghani**[1,5]*

**1** Faculty of Medicine, Dept of Internal Medicine, Universiti Teknologi MARA (UiTM), Sg Buloh, Selangor, Malaysia, **2** Faculty of Medicine, Dept of Radiology, Universiti Teknologi MARA (UiTM), Sg Buloh, Selangor, Malaysia, **3** Faculty of Medicine, Dept of Pathology, Universiti Teknologi MARA (UiTM), Sg Buloh, Selangor, Malaysia, **4** Faculty of Medicine, Dept of Public Health, Universiti Teknologi MARA (UiTM), Sg Buloh, Selangor, Malaysia, **5** Laboratory and Forensic Medicine (I-PPerForM), Institute of Pathology, Sg Buloh Campus, Universiti Teknologi MARA (UiTM), Selangor, Malaysia

* agrohana@gmail.com

## Abstract

### Introduction

There is limited data on the relationship between Obstructive Sleep Apnea (OSA) and Non-Alcoholic Fatty Liver Disease (NAFLD), each associated with increased cardiovascular risk. This study aimed to determine the relationships between severity of OSA, degree of steatosis in NAFLD and cardiovascular risk via CIMT and atherosclerosis markers ie intra-cellular adhesion molecule-1 (ICAM-1) an Lipoprotein-a (Lp(a)) in a group of patients with OSA.

### Materials and methods

This was a cross-sectional, single center study. A total of 110 subjects between 18 to 65 years of age and diagnosed with OSA following sleep study examinations were recruited. Exclusion criteria included seropositive Hepatitis B or Hepatitis C, and significant alcohol intake.

### Result

The prevalence of NAFLD was 81.8%. The mean CIMT (0.08±0.03 vs 0.06±0.01 cm, p = 0.001), ICAM-1 (334.53±72.86 vs 265.46±102.92 ng/mL, p = 0.001) and Lp(a) (85.41±52.56 vs 23.55±23.66 nmol/L, p<0.001) were significantly higher in the NAFLD group compared to the non-NAFLD group. Comparisons between the different groups showed significantly increasing levels of CIMT, ICAM-1 and Lp(a), lowest within the non-NAFLD, followed by the NAFLD 1 and NAFLD 2+3 groups. There was a significant positive correlation between degree of steatosis and the severity of OSA (r = 0.453, p<0.001). Logistic regression analysis revealed that patients with apnea/hypopnea index (AHI) of >30 were 52.77 (CI 6.34, 439.14) times more likely to have NAFLD compared to those with mild AHI (p<0.001).

**Data Availability Statement:** All relevant data are in the paper and its Supporting Information files.

**Funding:** This research received no specific grant from any funding agency in the public, commercial

or not-for-profit sectors. It was performed as part of the employment of the authors in Universiti Teknologi MARA (UiTM).

**Competing interests:** The authors have declared that no competing interests exist.

## Conclusion

The prevalence of NAFLD is alarmingly high in this group of OSA patients. The degree of steatosis in patients with NAFLD was significantly correlated with severity of OSA, CIMT measurements, ICAM-1 and Lp(a). Our findings underscore screening for NAFLD in patients with OSA to ensure prompt risk stratification and management.

## Introduction

The prevalence of obesity has gradually risen globally, with a doubling from 6.4% in 1980 to 12.0% in 2008 and currently poses significant public health problems [1,2]. The prevalence of obesity among Malaysians was estimated at 30.2% in 2015 [3], rising exponentially from what was reported by the Malaysian National Health Morbidity Survey in 1996, which then stood at 5.8% [4]. Obesity has been shown to be strongly associated with various non-communicable diseases (NCD), such as hypertension, type 2 diabetes, coronary heart disease, stroke, obstructive sleep apnea and non-alcoholic fatty liver disease (NAFLD), to list a few.

Obstructive sleep apnea (OSA) is a disorder that is characterized by recurrent upper airway collapse during sleep, leading to sleep fragmentation and daytime sleepiness with chronic intermittent hypoxia (CIH) [5]. It has been associated with a profound number of metabolic disorders including insulin resistance and dysglycemia, dyslipidemia, hypertension and subclinical atherosclerosis [6]. In recent years, despite the plethora of evidence to support the association between OSA and cardiovascular diseases, the void for more knowledge on the complexity of this anatomical and yet systemic illness remain apparent.

Non-alcoholic fatty liver disease (NAFLD) has been increasingly recognized as a liver disease that develops in the absence of alcohol abuse and imposes a major health burden. It is a progressive condition, whereby sustained liver injury could lead to further hepatocytes damage, and thus includes a spectrum of the disease severity, ranging from steatosis without inflammation to non-alcoholic steatohepatitis (NASH) and ultimately liver cirrhosis [7]. The pathophysiology is complex and related to oxidative stress and endothelial inflammation with the production of many pro-inflammatory cytokines including tumor necrosis factor alpha (TNF-α), interleukin-6 (IL-6), C-reactive protein (CRP) and interleukin-8 (IL-8) [8]. The interest in NAFLD has been increasing of late, with some reports on the clinical and histological features of NAFLD within the country [9,10]. Liver biopsy remains the gold standard to diagnose and categorize the severity of NAFLD. However, although it represents the best diagnostic tool for fatty liver, this procedure is invasive and could not be widely performed on apparently healthy subjects, mainly for ethical as well as practical reasons. Therefore, non-invasive tests and biomarkers of NAFLD are desirable alternatives. Liver ultrasonographic scanning has been demonstrated to have good correlations with histological findings of fatty infiltration, and has been universally accepted as a method for evaluation of severity of fatty liver [11].

The carotid artery intima-media thickness (CIMT) is a recognized tool to identify subclinical atherosclerotic disease [12]. Various cross-sectional studies have reported significantly greater carotid artery wall thickness in patients with NAFLD compared to those without NAFLD [13]. A large population-based study also suggests premature atherosclerosis in patients with NAFLD [14]. In regards to OSA, there is limited data on the measurements of CIMT to represent subclinical atherosclerosis in this group of patients.

Various surrogate markers are currently used in clinical studies to determine subclinical atherosclerosis, which include the Framingham Risk Score (accounting for age, gender,

hypertension, smoking, and hyperlipidemia status), carotid artery intima-media thickness (CIMT), high sensitivity C-reactive protein (hsCRP), atheroma formation, mediastinal fat pad, endothelial dysfunction and coronary calcium scores [15]. In addition, plasma levels of inter-cellular adhesion molecule-1 (ICAM-1) and E-selectin have been demonstrated as molecular markers for atherosclerosis and the development of coronary heart disease [16]. More recently, another marker that has been shown to predict subclinical atherosclerosis is lipoprotein a, Lp (a), which is a low density lipoprotein (LDL)-like particle synthesized within the liver hepato-cytes and then released into plasma [17]. A few studies have reported its role as an independent prospective risk factor for coronary heart disease attributed by its action on promoting athero-genesis and inhibiting fibrinolysis [18].

The relationship between OSA and NAFLD has been recognized in more recent years [19]. Benotti *et al* studied a group of subjects who were undergoing bariatric surgery and demon-strated an association between severity of OSA and histologically examined hepatocytic inflammation in patients with NAFLD, but was unable to reach statistical significance [20]. A few more studies have since provided more convincing evidence on the associations between the 2 conditions but these studies were limited by small sample sizes and mainly involving a specific group of patients undergoing bariatric surgery [21]. In view of the overlapping meta-bolic abnormalities, it may seem intuitive to assume that the co-existence of NAFLD and OSA would potentially increase cardiovascular risk further. In addition to further understanding that causal relationship between OSA and NAFLD, this study aimed to determine the down-stream effect of this double inflammatory cascades on atherosclerosis risk. To the best of our knowledge, to date, there has been no study on the direct associations between CIMT and the relevant atherosclerosis markers particularly ICAM-1 and Lp(a) in patients with OSA and ultrasound- proven NAFLD.

## Materials and methods

This was a cross-sectional study, conducted at the facilities of the Faculty of Medicine, Univer-siti Teknologi MARA, Malaysia. We screened 110 subjects consecutively, between the ages of 18 to 65, who were diagnosed with OSA as confirmed by sleep study examinations ie polysom-nograpy (PSG). They were subsequently categorized by the severity of their Apnea/ Hypopnea Index (AHI) ie mild AHI 5–15, moderate AHI 15–30, and severe AHI > 30/hr based on the polysomnography report [22]. We excluded patients with seropositive Hepatitis B or Hepatitis C, and had significant alcohol intake of more than 21 units per week in men and 14 units for women. The sampling method used was convenience sampling. Patients who fulfilled the inclusion criteria were approached, explained and informed regarding the main objectives of the study and blood samples were subsequently taken, prior to the initiation of positive airway pressure, if required. Confidentiality was assured to the respondents. A written consent was signed by respondents before proceeding with the study. Patients were allowed to opt out of the study at any time should they wish to. The study was conducted in accordance to the Dec-laration of Helsinki. This study was approved by the Research Ethics Committee (REC) of Universiti Teknologi MARA (UiTM) REF: 600-IRMI (5/1/6).

All patients had an additional 5 mls of venous blood drawn following an overnight fast, prior to routine clinic appointments. Serum samples were separated from blood following cen-trifugation and stored at -20$^{\circ}$C until further analysis. Biochemical measurements were per-formed on an automated analytical platform (c501, Roche Diagnostics, Germany) according to standard procedures at an ISO 15189 accredited laboratory. ICAM-1 was performed using the ELISA technique and read with microplate reader, whilst Lp(a) was read using the immu-noassay analyzer. We refer to a previous study which has determined a clinically acceptable

reference range for ICAM-1 was 128.9–347.48 ng/ml, thus taking values above this to be of clinical significance [23]. In regards to Lp(a), we refer to the report by Nordestgaard et al, which recommended that a value of approximately 50 mg/dL should be the desired level for reduction of cardiovascular risk [24].

Abdominal ultrasound examinations were performed by 2 independent radiologists. Subjects were scanned in the supine position by two independent radiologists using a high frequency 7.5 MHz linear array transducer using Philips iU22 imaging system. All measurements were made at the time of the scan on frozen images of longitudinal scans by using the machine's electronic caliper. Evidence of NAFLD was confirmed with radiological technique of liver-kidney contrast on degree of steatosis and further divided into four grades, severe (NAFLD-3), moderate (NAFLD-2), mild (NAFLD-1) and normal (non-NAFLD).

For the CIMT measurements subjects were also scanned in the supine position by two independent radiologists utilizing the same equipment. The distance between the 2 lines gave a reliable index of the thickness of the carotid intimal-medial complex. All measurements were made at the time of the scan on frozen images of longitudinal scans by using the machine's electronic caliper. Carotid segments for far (posterior) walls of each common carotid artery at a distance of 1cm from the bulb will be examined. The average of right and left CIMT were calculated and were recorded into millimeters (mm). The abnormal CIMT was defined as measurement of above 0.80 mm [25].

## Statistical analysis

Statistical Package for the Social Sciences (SPSS for Windows Version 22.0, SPSS Inc., Chicago, IL, USA) was used for all statistical analysis. Data on patients' sociodemographic information and characteristics are presented as mean and standard deviations for parametric data. For non-parametric data, the results are presented with median and interquartile range. Categorical data are presented as numbers of patients and percentages.

Comparisons between NAFLD and non-NAFLD was analyzed using Chi-Squared tests for categorical variables and independent- t test for continuous variables. Comparison for different degrees of steatosis in NAFLD, namely grades 0 (no steatosis), 1 (mild), 2 (moderate) and 3 (severe) groups were analyzed using chi-square test for categorical variables and ANOVA test for continuous variables. Correlation between 2 continuous variables was analyzed using Pearson rho test for normally distributed data and Spearman's rho test for non-parametric data. Simple linear regression was done for variables which are significantly correlated. Significance level was set at $p<0.05$. Univariate analysis was performed using linear regression and analysis of variance (ANOVA) to identify potential associated factors. Correlation between two continuous variables was analyzed with Spearman's rho test for non-normally distributed data. Cohen's (1988) cut-off points for interpretation of correlation strength are used [26]. A $p$ value $<0.05$ is considered significant.

## Results

A total of 110 patients with OSA were recruited. The baseline characteristics are presented in (**Table 1**). The mean age of the study population was 50.11±13.91 years. The majority of them were males (65.5%) and of Malay ethnicity (78.2%). Most of the participants were non-smokers (61.8%) and obese (72.7%). The mean waist circumference was 106.17±17.53 cm and the mean for hip circumference was 110.78 ±15.01 cm. Hypertension and dyslipidemia were diagnosed in 63.6% (n = 70) and 56.4% (n = 62) of the study population, respectively. However, most of the patients had reasonably good blood pressure control with mean systolic blood pressure (SBP) of 140.72 ±17.78 mmHg and mean diastolic blood pressure (DBP) of

**Table 1. Sociodemographic and baseline characteristics of overall study population.**

| Variables | N (%) | Mean±SD |
|---|---|---|
| Age | | 50.11±13.91 |
| Gender | | |
| Male | 72 (65.5%) | |
| Female | 38 (34.5%) | |
| Ethnicity | | |
| Malay | 86 (78.2%) | |
| Chinese | 9 (8.2%) | |
| Indian & *Others | 15 (13.6%) | |
| Comorbidities | | |
| Hypertension: | 70 (63.6%) | |
| Diabetes mellitus | 50 (45.5%) | |
| Dyslipidemia | 62 (56.4%) | |
| Ischemic Heart Disease | 7 (6.4%) | |
| **Others | 7 (6.4%) | |
| Smoking Status | | |
| Never | 68 (61.8%) | |
| Current | 17 (15.5%) | |
| Former | 25 (22.7%) | |
| Weight mean±SD, kg | | 91.12±22.36 |
| Height mean±SD, cm | | 162.76±10.40 |
| BMI mean±SD, kg/m2 | | 34.67±9.44 |
| BMI | | |
| Obese (≥27.5) | 80 (72.7%) | |
| Overweight (23–27.4) | 18 (16.4%) | |
| Normal (≤22.9) | 12 (10.9%) | |
| Waist Circumference mean±SD cm | | 106.17±17.53 |
| Hip Circumference mean±SD, cm | | 110.78±15.01 |
| Waist hip ratio | | 0.95±0.08 |
| Systolic blood pressure, mean±SD, mmHg | | 140.72±17.78 |
| Diastolic blood pressure mean±SD, mmHg | | 81.03±12.26 |
| Mean Apnea Hypopnea Index (AHI) | | |
| Severity of Obstructive Sleep Apnea (AHI), | | |
| Mild (5–15) | | |
| Moderate (16–30) | 27 (24.5%) | |
| Severe (>30) | 33 (30.0%) | |
| | 50 (45.5%) | |
| NAFLD status | | |
| Yes | 90 (81.8%) | |
| No | 20 (18.2%) | |
| Degree of steatosis | | |
| Normal | 20 (18.2%) | |
| Grade 1 | 47 (42.7%) | |
| Grade 2 | 42 (38.2%) | |
| Grade 3 | 1 (0.9%) | |
| Carotid Intima Media thickness mean (SD) mm | | 0.80±0.20 |
| ≥ 0.80 mm | 54 (49.1%) | |
| < 0.80 mm | 56 (50.9%) | |

(*Continued*)

**Table 1.** (Continued)

| Variables | N (%) | Mean±SD |
|---|---|---|
| ALT mean±SD, U/L | | 32.14±23.67 |
| ALP mean±SD, U/L | | 78.71±23.31 |
| AST mean±SD, U/L | | 21.87±16.16 |
| GGT mean±SD, U/L | | 53.39±57.02 |
| Intracellular Adhesion Molecule-1, mean±SD, ng/ml | | 321.97±83.05 |
| $\geq$ 347.48 ng/ml | 70 (63.6%) | |
| < 347.48 ng.ml | 40 (36.4%) | |
| Lipoprotein (a), mean±SD, nmol/L | | 74.16±54.11 |
| $\geq$ 75 nmol/L | 56 (50.9%) | |
| < 75 nmol/L | 54 (49.1%) | |
| HbA1c (n = 50), mean±SD, % | | 7.75±1.69 |

81.03±12.26 mmHg. Type 2 diabetes mellitus (T2DM) was present in 50 subjects (45.5%), with good glycemic control, mean HbA1c of 7.75 ±1.69%, and almost all of them (90%) were on metformin. In regards to the OSA status, almost half (45.5%) of the study population had severe AHI of more than 30. The liver enzymes were within normal range. The mean values for both intercellular adhesion molecule -1 (ICAM-1) and lipoprotein (a) [Lp(a)] were within the low risk ranges, ie 321.97±83.05 ng/ml and 74.16±54.11 nmol/L, respectively. However, high levels were detected in at least half of study population, with ICAM-1 level of $\geq$347.48 ng/ml observed in 63.6% and high level of Lp(a) of $\geq$75 nmol/L seen in 50.9%. Finally, mean CIMT in this study population was 0.80±0.20 mm. Refer **Table 1**.

The prevalence of NAFLD within the study population was notably high at 81.8% (95% CI: 74.5, 89.1) (n = 90). Mean weight in the NAFLD group was significantly higher compared to the non-NAFLD group (94.77±21.85 kg vs 74.67±16.80 kg, p < 0.001), with 82.2% of the NAFLD group being obese (p<0.001). The NAFLD group also had significantly higher mean SBP and liver transaminases. Both the atherosclerosis markers ie mean ICAM-1 and Lp(a) were significantly higher in the NAFLD group compared to the non-NAFLD group (334.53 ±72.86 vs 265.46 ±102.92 ng/mL, p = 0.001, 85.41±52.56 vs 23.55±23.66 nmol/L, p <0.001, respectively). Subsequent categorical analyses showed higher percentage of patients with high ICAM level of >347.48 ng/ml and Lp(a) >75 nmol/L within the NAFLD group compared to the non-NAFLD group, (56.7% vs 5%, p <0.001 and 58.9% vs 1.9%, p <0.001, respectively). **Table 2**. Mean CIMT was higher in the NAFLD group compared to the non-NAFLD group (0.80±0.30 vs 0.60±0.10 mm, p = 0.001). The clinically significant abnormal CIMT of >0.8 mm was detected in 64.4% of patients with NAFLD compared to only 5% within the non-NAFLD group, p <0.002. Refer **Table 2**.

We further categorized the NAFLD group according to the severity of the steatosis, with 52% within grade 1 steatosis (n = 47), 46% within grade 2 steatosis (n = 42) and 2% had grade 3 steatosis (n = 1). Grades 2 and 3 were subsequently combined for analyses. There were significant differences in SBP, DBP and liver transaminases between non-NAFLD, NAFLD 1 and NAFLD 2+3 groups. CIMT measurements were significantly highest in the NAFLD 2+3 group, followed by NAFLD 1, compared to the non-NAFLD groups (0.90±0.20, 0.70±0.30, 0.60±010 mm, respectively, p<0.001). Levels of ICAM-1 and Lp(a) were also seen to be significantly highest in the NAFLD 2+3 group, followed by the NAFLD 1 group as compared to the non-NAFLD. Refer **Table 3**.

From the overall study population, 24.5% (n = 27) had mild AHI, 30% (n = 33) had moderate AHI and 45.5% (n = 50) had severe AHI. It was interesting to note that there was a

**Table 2. Comparison between non-NAFLD and NAFLD group.**

| Variables | NAFLD (N = 90) | Non-NAFLD (N = 20) | p-value |
|---|---|---|---|
| Systolic blood pressure mean±SD, mmHg | 143.23±16.33 | 129.35±19.96 | 0.001 |
| Diastolic blood pressure mean±SD, mmHg | 82.08±11.95 | 76.30±12.81 | 0.056 |
| ALT mean±SD, U/L | 34.93±25.05 | 19.57±8.47 | 0.008 |
| ALP mean±SD, U/L | 79.68±23.74 | 74.35±21.30 | 0.358 |
| AST mean±SD, U/L | 34.27±16.72 | 21.05±6.11 | 0.001 |
| GGT mean±SD, U/L | 59.10±61.26 | 27.70±15.12 | 0.025 |
| Severity of OSA (AHI), n (%) | | | |
| Mild (5–15) | 13 (14.4%) | 14 (70.0%) | <0.001 |
| Moderate (16–30) | 28 (31.1%) | 5 (25.0%) | |
| Severe (>30) | 49 (54.4%) | 1 (5.0%) | |
| Intracellular Adhesion Molecule-1, mean±SD, ng/ml | 334.53 ±72.86 | 265.46 ± 102.92 | 0.001 |
| ≥ 347.48 ng/ml | 39 (56.7%) | 1 (5%) | |
| < 347.48 ng.ml | 51 (43.3%) | 19 (95%) | 0.001 |
| Lipoprotein (a), mean±SD, nmol/L | 85.41±52.56 | 23.55±23.66 | <0.001 |
| ≥ 75 nmol/L | 53 (58.9%) | 5 (1.9%) | <0.001 |
| < 75 nmol/L | 37 (41.1%) | 95 (33.9%) | |
| Carotid Intima Media thickness mean±SD, mm | 0.80±0.30 | 0.60±0.10 | <0.001 |
| ≥ 0.80 mm | 58 (64.4%) | 1 (5%) | 0.002 |
| < 0.80 mm | 32 (35.6%) | 19 (95%) | |

significantly higher proportion of patients with NAFLD 2+ 3 within the severe AHI group compared to the mild AHI group (61.9% vs 14.3%, p<0.001). There was indeed a significant correlation between the severity of OSA and NAFLD; r = 0.384, p<0.001. Refer **Table 4.**

Logistic regression analysis was performed to ascertain the effects of blood pressure, BMI, glycemia, severity of OSA, waist circumference, hip circumference, ICAM-1 and Lp(a) on the likelihood of OSA patients having NAFLD. Patients with OSA and elevated AHI of >30 were 52.77 times more likely to have NAFLD compared to those with mild AHI (5–15) (p < 0.001). Patients with OSA were also more likely to have NAFLD with co-existing hypertension (OR 4.33, p = 0.005; diabetes mellitus (OR 3.0, p = 0.049; and dyslipidemia (OR 2.92, p = 0.038). In addition, elevated surrogate markers of ICAM-1 and Lp(a) level were more likely to be present in patients with NAFLD with OR 14.53 and 27.22, respectively (p = 0.011 and p = 0.001). Refer **Table 5**.

Multiple logistic regression analysis was performed to adjust for the predictors for NAFLD in OSA patients. The interaction and multicollinearity were also checked within the overall

**Table 3. Comparison of clinical parameters between different groups of NO NAFLD, NAFLD 1 and NAFLD 2+3.**

| Variables | Non-NAFLD (N = 20) | NAFLD Grade 1 (N = 47) | NAFLD Grades 2 +3 (N = 43) | p-value |
|---|---|---|---|---|
| Systolic Blood Pressure, mean±SD, mmHg | 129.35±19.96 | 138.55±16.43 | 148.26±14.95 | <0.001 |
| Diastolic Blood Pressure, mean±SD, mmHg | 76.30±12.81 | 79.77±10.45 | 84.67±12.21 | 0.027 |
| ALP mean±SD, U/L | 74.35±21.30 | 79.57±23.58 | 79.59±23.44 | 0.669 |
| ALT mean±SD, U/L | 19.56±8.47 | 30.70±22.00 | 39.00±27.72 | 0.008 |
| AST mean±SD, U/L) | 21.06±6.12 | 31.18±12.37 | 37.00±19.85 | 0.001 |
| GGT mean±SD, U/L) | 27.70±15.12 | 59.30±73.79 | 58.93±45.12 | 0.085 |
| Intracellular Adhesion Molecule-1, mean±SD, ng/ml | 265.46±102.92 | 333.74±78.00 | 334.16±68.04 | 0.003 |
| Lipoprotien (a), mean±SD, nmol/L | 23.55±23.70 | 79.30±56.07 | 92.21±48.80 | <0.001 |
| CIMT mean±SD, mm | 0.60±0.10 | 0.70±0.30 | 0.90±0.20 | <0.001 |

**Table 4. The correlation between severity of NAFLD and OSA (AHI).**

| OSA Classification | NAFLD Classification | | | p-value for association[a] | r | p-value for correlation[b] |
|---|---|---|---|---|---|---|
| | Normal | NAFLD Grade 1 | NAFLD Grade 2 | | | |
| Mild | 14 (70.0) | 7 (14.9) | 6 (14.3) | <0.001* | 0.384 | <0.001* |
| Moderate | 5 (25.0) | 18 (38.3) | 10 (23.8) | | | |
| Severe | 1 (5.0) | 22 (46.8) | 26 (61.9) | | | |

Notes

* Statistically significant at α = 0.05.

Statistical test

[a] chi-square test

[b] Kendal tau'b.

study cohort. All the significant factors in Table 5 were included in this analysis. Notably, the small number in the non-NAFLD group limit the statistical power. Nevertheless, two factors were subsequently identified to be significant predictors for the detection of NAFLD in OSA patients, which were ICAM-1 level (p = 0.002) and Lp(a) (p = 0.003). **Table 6.** Elevated ICAM-1 level of ≥347ng/ml demonstrated an OR of 41.69; CI, 3.76–461.97; p = 0.002, whilst high Lp(a) level of ≥75 nmol/L showed an OR of 35.12; CI, 3.47–355.40; p = 0.003) for NAFLD. The Receiver Operative Characteristic (ROC) curve of these factors associated with NAFLD showed an area under the curve of 95.6% (95%CI: 91.9, 99.3). It could, therefore, be inferred that OSA patients who had elevated levels of ICAM-1 and Lp(a) are highly likely to have NAFLD (**Fig 1**).

## Discussion

The prevalence of NAFLD in this group of OSA patients was notably high (81.8%, n = 90/110). We observed that this number is similar to a study by Turkay *et al* who reported a prevalence of 71.2% [27]. The most probable reason for this similarity is the high number of obese subjects within both study populations. Over the last decade, there has been increasing interest in understanding the associations between OSA and NAFLD. A recent review on OSA and NAFLD summarized that OSA is indeed associated with increased liver fibrosis and raises a question on the need for screening of these conditions, followed by aggressive therapy to reduce future cardiovascular risks [21]. The association between the high prevalence of NAFLD in OSA patients has been suggested to be attributed to the pathogenesis of fatty liver disease caused by continuous intermittent hypoxia (CIH) that lead to sympathetic activation, endothelial dysfunction, oxidative stress and systemic inflammation [28]. However, data from a cross-sectional study by Norman *et al* have further recommended that CIH is in fact an independent causal factor for the development of NAFLD [29]. We therefore concur that perhaps there is a direct association between AHI and NAFLD, suggesting some effect of the severity of apnea on hepatic steatosis, albeit not as a significant independent predictor for the latter based on the multivariate analysis.

An important aspect of this study is the evidence of subclinical atherosclerosis in this particular cohort of patients with OSA. A study by Targher *et al* had shown a markedly greater CIMT than control subjects (1.14 ± 0.20 vs. 0.82 ± 0.12 mm; P < 0.001) [30]. Another study by Silvestrini *et al* similarly showed that CIMT in OSA patients was significantly higher than that of control subjects (1.429 ± 0.34 vs 0.976 ± 0.17 mm, p <0.0001) [31]. However, there is paucity of data on the cumulative atherosclerotic risk of patients with co-existing OSA and NAFLD. Petta et al, showed that in a group of patients with NAFLD who subsequently underwent cardiorespiratory polygraphy to detect OSA, the prevalence of significant thickening of

**Table 5. Analysis of associated factors for NAFLD in OSA patients.**

| | OR (95%CI) | p-value |
|---|---|---|
| Age | 1.03 (0.99, 1.07) | 0.071 |
| Gender: | | |
| Male | 1.72 (0.64, 4.81) | 0.280 |
| Female | 1 | Ref |
| Race: | | |
| Malay | 1.29 (0.32, 5.16) | 0.723 |
| Chinese | 0.50 (0.08, 3.27) | 0.469 |
| Indian & Others | 1 | Ref |
| Hypertension: | | |
| Yes | 4.33 (1.56, 12.06) | 0.005 |
| No | 1 | Ref |
| DM: | | |
| Yes | 3.00 (1.01, 8.95) | 0.049 |
| No | 1 | Ref |
| Dyslipidemia: | | |
| Yes | 2.92 (1.06, 8.03) | 0.038 |
| No | 1 | Ref |
| Ischemic Heart Disease: | | |
| Yes | 1.62 (0.43, 6.09) | 0.476 |
| No | 1 | Ref |
| IHD family: | | |
| Yes | 2.29 (0.77, 6.86) | 0.137 |
| No | 1 | Ref |
| Smoking: | | |
| Never | 1 | Ref |
| Current | 0.92 (0.26, 3.24) | 0.896 |
| Former | 6.79 (0.85, 54.42) | 0.071 |
| BMI: | | |
| Obese | 17.27 (4.18, 71.25) | <0.001 |
| Overweight | 2.20 (0.50, 9.75) | 0.229 |
| Normal | 1 | ref |
| Severity of OSA: | | |
| Mild | 2 | Ref |
| Moderate | 6.03 (1.79, 20.32) | 0.004 |
| Severe | 52.77 (6.34, 439.14) | <0.001 |
| Waist | 1.08 (1.04, 1.12) | <0.001 |
| Hip | 1.07 (1.03, 1.12) | 0.001 |
| SBP | 1.05 (1.02, 1.09) | 0.003 |
| DBP | 1.04 (0.99, 1.09) | 0.060 |
| ICAM-1 | | |
| $\geq$ 347 ng/ml | 14.53 (1.86, 113.28) | 0.011 |
| < 347 ng/ml | 1 | ref |
| Lp(a) | | |
| $\geq$ 75 nmol/L | 27.22 (3.49, 212.31) | 0.002 |
| < 75 nmol/L | 1 | ref |
| HbA1c (n = 50) | 1.21 (0.95, 5.15) | 0.067 |

**Table 6. Multivariate analysis in overall cohort.**

| | Adj. OR (95%CI) | p-value |
|---|---|---|
| ICAM-1 | | |
| ≥ 347ng/ml | 41.69 (3.76, 461.97) | 0.002 |
| > 347ng/ml | 1 | |
| Lp(a) | | |
| ≥ 75 nmol/L | 35.12 (3.47, 355.40) | 0.003 |
| > 75 nmol/L | 1 | |

Notes

Sensitivity: 94.4; Specificity: 75.0.

Model fit (p>0.05), Hosmer and Lemeshow: 0.967.

Cox & Snell: 41.3.

Forward Method.

No multicollinearity and interaction problem.

CIMT was higher in patients at high risk compared to those at low risk for OSA (51.2% vs 24.3%, p = 0.006) [32]. However, the study only employed the use of the STOP-BANG questionnaire to identify risk for OSA whilst our study confirmed its diagnosis. Moreover, we demonstrated that the mean CIMT values were highest in those within the severe NAFLD grades 2

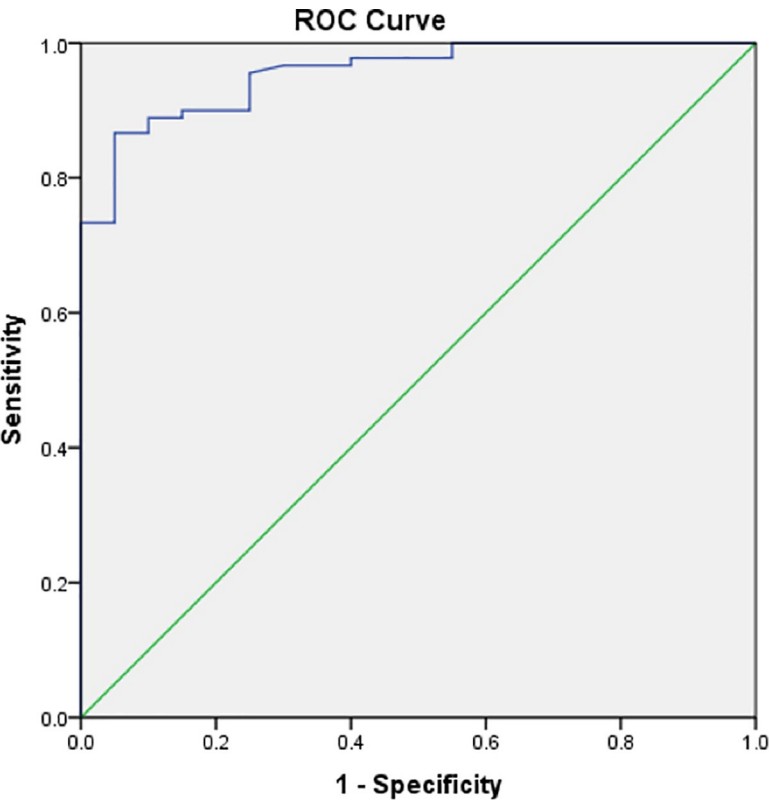

**Fig 1. ROC curve of the two significant factors associated with having NAFLD.** The area under the curve is 95.6% (95%CI: 91.9, 99.3).

and 3 compared to NAFLD 1 and non-NAFLD. There were statistically significant positive correlations between CIMT and SBP (r = 0.233, p = 0.014) among OSA patients as well as in NAFLD (r = 0.217, p = 0.040). This suggests the importance of good SBP control in patients with OSA and NAFLD.

ICAM-1 is proven to be a molecular marker for atherosclerosis and the development of coronary heart disease [33]. Our data is in agreement with previous studies including a meta-analysis which included 51 studies and concluded that ICAM-1 was indeed higher in patients with OSA compared to control group [34]. In this present study we demonstrated that more than half of the study population (63.6%) had high ICAM-1 level (> 347.48 ng/ml). Deneva-Koyvheva *et al* recommended the clinically acceptable reference interval for ICAM-1 was 128.9–347.48 ng/ml [22]. The mean ICAM-1 level was significantly higher within the NAFLD compared to the non-NAFLD group, which was consistent with previous studies [35,36]. Earlier studies reported a relationship between ICAM-1 and CIMT as subclinical atherosclerosis risk, which concluded that ICAM-1is independent of other known CHD risk factors and is strongly associated with carotid plaques [37–39]. Consequently, we also demonstrated that higher levels of ICAM-1 corresponded to worsening severity of steatosis, which underscores the escalating cardiovascular risk in these patients at the end of the spectrum of the disease.

Lipoprotrien (a) [Lp(a)] is a low density lipoprotein (LDL)-like particle and has been recognized as another inflammatory marker predicting subclinical atherosclerosis [23]. More recently, the Dallas Heart Study showed Lp(a) as a positive predictor for major cardiovascular event (MACE) with hazard ratio of 2.35 [40]. Interestingly our study demonstrated that half of the patients with OSA (50.9%) had significantly high levels of Lp(a) of above 75 nmol/L, of which this Lp(a) cut-off level is regarded as having high atherosclerosis risk based on the evaluation of the Framingham data [23]. High serum Lp(a) levels correlate with premature atherosclerosis and stroke with an an approximate doubling of coronary risk when the Lp(a) level rises above 75 nmol/L, with the risk further escalating to approximately 6-fold in the addition of a high LDL-cholesterol [41,42]. To the best of our knowledge, there has been no study to determine the association between the Lp(a) with the severity of OSA as well as the NAFLD populations. There were significant differences in Lp(a), being highest in NAFLD grade 2 group as compared to NAFLD-1 and non-NAFLD groups (92.21 ± 48.80 vs 79.30 ± 56.07 vs 23.55 ± 23.70 nmol/L, p < 0.001). Thus, our data further supports the recommendation by the European Atherosclerosis Society that Lp(a) measurement should be recommended in patients with high cardiovascular risk including family history of premature cardiovascular disease [42] and subsequently those with OSA and NAFLD.

The multivariate analyses demonstrated a few novel points worth noting. Patients with severe OSA as defined by elevated AHI of >30 were 52 times more likely to have NAFLD compared to those with mild AHI, which were further worsened by other comorbidities including hypertension, diabetes mellitus and dyslipidemia. In addition, elevated inflammatory markers of ICAM-1 and Lp(a) are demonstrated to be significant predictors for the detection of NAFLD in OSA patients. These findings point to the understanding that patients with OSA and co-existing NAFLD are at an alarmingly high risk of cardiovascular events or disease. It also suggests that, in addition to other conventional risk factors assessment, these patients should have further tests to stratify their cardiovascular risks among which is the use of these 2 surrogate markers, ICAM-1 and Lp(a).

To date, there is scarcity in the number of studies evaluating the relationship between the OSA severity via AHI with NAFLD. Nonetheless, this study reported a similar finding to Turkay et al, who demonstrated that as NAFLD severity increased from mild to severe form, mean AHI and oxygen desaturation index values were also significantly increased [26]. Conversely, our study showed that increasing severity of OSA, based on AHI values, correlated with

severity of NAFLD based on the grades 0–3, r = 0.453, p<0.001. The causal or effect relationship remains vague but suffice to note the synergistic consequence of the co-existing insults.

Comparisons between NAFLD and non-NAFLD groups demonstrated statistically significant higher weight, BMI, systolic blood pressure, waist circumference, hip circumference and waist to hip ratio within the former group. Obesity is a co-existential factor between the 2 conditions and plays an important causative factor for both. Ong *et al* observed that the prevalence of NAFLD could be as high as 93% among morbidly obese patients and 9% of them had advanced fibrosis (i.e., bridging fibrosis or cirrhosis [43]. Interestingly, our study demonstrated similar finding as 92.5% of our NAFLD and OSA patients were obese. Other co-morbidities including dyslipidemia, hypertension and diabetes mellitus are also well recognized associated factors for NAFLD. However, we emphasize the importance of these co-morbidities on the likelihood of developing NAFLD. OSA patients with high SBP, expanded waist and hip circumferences were more likely to have NAFLD compared to leaner patients. This could add a very important impact on screening for NAFLD in a subgroup of patients who have been diagnosed with severe OSA, and found to be obese, with uncontrolled blood pressure, to prevent major adverse cardiac events and death.

Our data seems to be quite consistent with similar studies within the Asian region, whereby, the prevalence of ultrasound-diagnosed NAFLD among OSA patients has been reported to range between 60% and 90% [44–46]. However, we acknowledge that this is probably an overestimation in comparison to a histopathological diagnosis of NAFLD, which could be much lower in patients with histology-proven NAFLD [47]. Nonetheless, the risk remains high as demonstrated by a cross-sectional study by Musso *et al* (n = 2,183) which showed that patients with OSA had 2.01 times (95% CI: 1.36–2.97) higher risk to have NAFLD confirmed by histopathological evidence [48]. Other imaging modalities have emerged as diagnostic tools for NAFLD in recent times. From a meta-analysis involving 46 studies, which compared the sensitivity and specificity of various imaging modalities including ultrasonography, computed tomography (CT) scan, magnetic resonance imaging (MRI) and proton magnetic resonance spectroscopy (1H-MRS), the authors concluded that the latter 2 newer techniques were acceptable in the evaluation of hepatic steatosis [49]. In due course, MRI has become increasingly popular in establishing the diagnosis of NAFLD mainly attributed to the availability of quantification of hepatic steatosis by measuring proton density fat fraction (PDFF) [50]. Its use, however, is limited by cost and accessibility. Therefore, liver ultrasonography scanning remains the radiological imaging of choice by virtue of its non-invasiveness and convenience. Furthermore, it has been shown to have a good correlation with histological findings of fatty infiltration, and has been accepted worldwide as a method of evaluation for the different degrees of fatty liver [7,51], with an accuracy of 88% in direct comparison with histopathological findings [52].

We acknowledge the limitations of this study such as the single center data collection, the cross-sectional design and the relatively small number of subjects. Furthermore, a comparison with a group of healthy subjects without OSA would have been ideal. The number of subjects with high degree of steatosis ie grade 3 was very small and made sub-group analysis difficult. The use of fibroscan to assess the degree of steatosis would also be interesting to be incorporated as a research tool. The exclusion of viral or autoimmune hepatitis were obtained based on medical records and patients' disclosures only and not from actual serological testing. Future studies to address these limitations would help strengthen the findings from this study.

## Conclusion

In conclusion, this study revealed a high prevalence of NAFLD within a group of OSA patients. We were able to demonstrate that severity of OSA, categorized by the AHI, was significantly

associated with severity of NAFLD. Subclinical atherosclerosis, represented by abnormal levels of CIMT, ICAM-1 and Lp(a), were significantly predominant among NAFLD subjects. ICAM-1 and Lp(a) were strong predictors for detection of NAFLD in the study cohort. This study highlighted the elevated cardiovascular risk of patients with severe OSA and NAFLD, the importance of early screening and detection of this chronic inflammatory condition. It also stresses the importance of optimizing cardiovascular risk factors in this group of high-risk patients as part of a preventive management strategy to avoid the consequential debilitating and fatal cardiovascular events.

## Supporting information

**S1 File.**
(SAV)

## Acknowledgments

We would like to thank all the medical and non-medical personnel involved in the data collection and patient management.

## Author Contributions

**Conceptualization:** Fatimah Zaherah Mohamed Shah, Rohana Abdul Ghani.

**Data curation:** Samshol Sukahri, Marymol Koshy, Bushra Johari, Mazuin Mohd Razali, Thuhairah Hasrah Abdul Rahman.

**Formal analysis:** Mohamad Rodi Isa.

**Investigation:** Marymol Koshy, Thuhairah Hasrah Abdul Rahman.

**Methodology:** Fatimah Zaherah Mohamed Shah, Rohana Abdul Ghani.

**Software:** Bushra Johari.

**Supervision:** Ahmad Izuanuddin Ismail.

**Writing – original draft:** Rohana Abdul Ghani.

**Writing – review & editing:** Mazuin Mohd Razali, Thuhairah Hasrah Abdul Rahman, Mohamad Rodi Isa, Rohana Abdul Ghani.

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
