## [Decision Letter · Decision Letter 0]

8 Apr 2021

PONE-D-21-03573

Significantly Higher Atherosclerosis Risks in Patients with Obstructive Sleep Apnoea and Non Alcoholic Fatty Liver Disease

PLOS ONE

Dear Dr. Abdul Ghani,

Thank you for submitting your manuscript to PLOS ONE. After careful consideration, we feel that it has merit but does not fully meet PLOS ONE’s publication criteria as it currently stands. Therefore, we invite you to submit a revised version of the manuscript that addresses the points raised during the review process.

Both reviewers had significant concerns that are outlined in their reviews.  Please make sure you respond carefully to each comment.  

We look forward to receiving your revised manuscript.

Kind regards,

James Andrew Rowley

Academic Editor

PLOS ONE

Journal Requirements:

1 .Please ensure that your manuscript meets PLOS ONE's style requirements, including those for file naming. The PLOS ONE style templates can be found at

Reviewers' comments:

Reviewer's Responses to Questions

**Comments to the Author**

1. Is the manuscript technically sound, and do the data support the conclusions?

Reviewer #1: Partly

Reviewer #2: Partly

2. Has the statistical analysis been performed appropriately and rigorously? 

Reviewer #1: Yes

Reviewer #2: No

3. Have the authors made all data underlying the findings in their manuscript fully available?

Reviewer #1: No

Reviewer #2: No

4. Is the manuscript presented in an intelligible fashion and written in standard English?

Reviewer #1: Yes

Reviewer #2: Yes

5. Review Comments to the Author

Reviewer #1: Degree of steatosis is not the same as stage/grade of NAFLD. Grade is determined by degree of inflammation (NAS score) and stage by fibrosis on biopsy (gold standard). If biopsy is not available severity can be assessed by specialized non invasive markers and scores like Fibroscan, NALFD fibrosis score etc. Hence in this paper the term grade/stage of NAFLD should be replaced by degree of steatosis.

It would be useful if the authors had any such surrogate markers of NAFLD severity and study the relationship of NAFLD severity with atherosclerosis markers.

What is the significance of table 5? It would seem that the authors are saying that in patients who do not have NAFLD, CIMT is not related to any metabolic or cardiovascular factor which does not seem to be true clinically. I think there are too few patients in Non NAFLD group to such correlation analysis in this subgroup.

Table 6 and table 7 are almost duplicate. One of them can be removed.

How many parameters were entered into the multivariate analysis model? With only 20 patients without NAFLD in the study, doing a multivariate analysis with many dependent variables will severely limit the power of the calculations.

Which parameters were studied in the multivariate analysis? It is surprising to note that BMI/obesity did not come out as independent predictors of NAFLD in this study. were these parameters included in the multivariate analysis?

"Inversely it could, therefore, be inferred that these OSA patients who had NAFLD had a 95.6% possibility of having elevated atherosclerosis risk as suggested by elevated levels of ICAM-1 and Lp(a)." I dont think this is a correct conclusion. AUROC of 95.6% for ICAM and LPa simply suggests that a predictive score using these 2 parameters together has 95.6% accuracy in predicting presence of NAFLD. To assess the probability of atherosclerotic risk in NAFLD one would have to calculate how many of NAFLD patients had elevated levels of one or more markers of atherosclerosis using acceptable cut-offs for each such marker and then calculate the atherosclerotic risk associated with each of those markers.

"However, there is paucity of data on the cumulative atherosclerotic risk of patients with co-existing OSA21

and NAFLD; and to the best of our knowledge this would be the first study utilizing CIMT and the inflammatory markers to demonstrate this." This is a misleading conclusion. Please see the point above.

"We concur with the findings by Petta et al, which showed that OSA was independently associated with subclinical carotid atherosclerosis as evaluated by CIMT in patients with NAFLD". The authors have not presented any such data showing association of CIMT with OSA, since there was no non OSA control group in this study to compare CIMT values with.

"Furthermore, our data demonstrated a significant rise in ICAM-1 levels with severity of OSA as categorized by

increasing AHI, from mild 259.16 ± 57.40 ng/mL, moderate 299.65 ± 90.41 ng/mL and severe 370.63 ± 57.71 ng/mL, p < 0.001". This data has not been presented anywhere in the results section.

The discussion is too long with many repetitions and can be condensed.

Reviewer #2: This was an interesting study that examined potential links between OSA, liver disease and CV risks. The authors studied 110 patients with suspected OSA. Authors found that OSA severity was significantly associated with liver disease. CV risk (CIMT) was elevated in liver patients. I had the following questions/comments:

1. In the discussion, other limitations that should be mentioned include: relatively small numbers, single centre, cross sectional nature

2. I wasnt sure if these were consecutive patients? if not, how were they selected?

3. When was the blood taken in relation to PSG (i.e. time frame)?

4. I assume none of these patients were treated?

5. Was there an analysis of desaturation indices (e.g. 4% ODI, mean saturation, % time below 90% etc). I would suspect these might be highly related to liver disease?

6. I wasnt sure why ICAM specifically was the inflammatory marker selected?

7. Mean BMI, AHI, saturation, etc should be added to Table 1. Did any patients have obesity hypoventialtion syndrome? This might have increased right sided pressures and lead to more liver swelling?

8. There should be more description of the ultrasound methods to detect liver disease. How validated are they in the context of substantial obesity +/- high right sided pressures?

9. In the last paragraph of page 8, do you mean carotid ultrasound or abdominal?

10. Was there an analysis of BMI, AHI as continuous variables? It was difficult to understand if Tables 7 and 8 reflected multivariate models? This should be made clearer.

11. The CI around the OR for severe osa is very wide-- these should be added to the abstract so readers understand the instability of these measures.

12. Was there a multivariate analysis looking at determinants of CIMT? I think you need this to determine whether liver disease is an independent risk factor.

6. PLOS authors have the option to publish the peer review history of their article (what does this mean?). If published, this will include your full peer review and any attached files.

Reviewer #1: **Yes: **Swastik Agrawal

Reviewer #2: No

---

## [Author Response · Author response to Decision Letter 0]

16 Apr 2021

We would like to thank both Reviewers for their constructive comments and suggestions. We hereby provide the responses to the questions and comments.

Reviewer #1: 

Degree of steatosis is not the same as stage/grade of NAFLD. Grade is determined by degree of inflammation (NAS score) and stage by fibrosis on biopsy (gold standard). If biopsy is not available severity can be assessed by specialized non invasive markers and scores like Fibroscan, NALFD fibrosis score etc. Hence in this paper the term grade/stage of NAFLD should be replaced by degree of steatosis.

Thank you for the clarification. This has been amended accordingly throughout the manuscript. Stages of NAFLD has been replaced as grades of steatosis Grade 0-4. 

It would be useful if the authors had any such surrogate markers of NAFLD severity and study the relationship of NAFLD severity with atherosclerosis markers.

Thank you for the suggestion. Unfortunately we do not have access to Fibroscan. This is a very good suggestion for us to continue with the study at a later opportunity. We have included this in the limitation as well.

What is the significance of table 5? It would seem that the authors are saying that in patients who do not have NAFLD, CIMT is not related to any metabolic or cardiovascular factor which does not seem to be true clinically. I think there are too few patients in Non NAFLD group to such correlation analysis in this subgroup.

The reviewer has highlighted a valid point. Table 5 has been removed as suggested. Table 6 has been adjusted to Table 5 accordingly.

Table 6 and table 7 are almost duplicate. One of them can be removed.

Thank you for highlighting that. We have removed Table 7. Table 8 has been adjusted to table 6 accordingly.

How many parameters were entered into the multivariate analysis model? With only 20 patients without NAFLD in the study, doing a multivariate analysis with many dependent variables will severely limit the power of the calculations.

Thank you for the comments. 10 parameters were included in the multivariate analysis as described in table 7 previously. We have acknowledged the small number of n=20 in the Non-NAFLD group in the result section. 

This sentence has been added : “All the variables in Table 5 were included in this analysis. Notably, the small number in the non-NAFLD group limit the statistical power. Nevertheless, two factors….”

Which parameters were studied in the multivariate analysis? It is surprising to note that BMI/obesity did not come out as independent predictors of NAFLD in this study. were these parameters included in the multivariate analysis?

Only the significant parameters were included in the multivariate analysis ie SBP, DBP, BMI, WC, ALP, ALT, AST, GGT, ICAM-1, Lp(a). 

Yes, BMI was included in the analysis, however it was not significant.

"Inversely it could, therefore, be inferred that these OSA patients who had NAFLD had a 95.6% possibility of having elevated atherosclerosis risk as suggested by elevated levels of ICAM-1 and Lp(a)." 

I don’t think this is a correct conclusion. AUROC of 95.6% for ICAM and LPa simply suggests that a predictive score using these 2 parameters together has 95.6% accuracy in predicting presence of NAFLD. To assess the probability of atherosclerotic risk in NAFLD one would have to calculate how many of NAFLD patients had elevated levels of one or more markers of atherosclerosis using acceptable cut-offs for each such marker and then calculate the atherosclerotic risk associated with each of those markers.

Thank you for the comment. The statement has been amended to include your comment. 

"It could, therefore, be inferred that OSA patients who had elevated levels of ICAM-1 and Lp(a) are highly likely to have NAFLD.”

"However, there is paucity of data on the cumulative atherosclerotic risk of patients with co-existing OSA

and NAFLD; and to the best of our knowledge this would be the first study utilizing CIMT and the inflammatory markers to demonstrate this." 

This is a misleading conclusion. Please see the point above.

Thank you for the comment. We have removed the statement of “……and to the best of our knowledge this would be the first study utilizing CIMT and the inflammatory markers to demonstrate this."

"We concur with the findings by Petta et al, which showed that OSA was independently associated with subclinical carotid atherosclerosis as evaluated by CIMT in patients with NAFLD". 

The authors have not presented any such data showing association of CIMT with OSA, since there was no non OSA control group in this study to compare CIMT values with.

Thank you for the comment. We have revised the statement to state the following: 

Petta et al, showed that in a group of patients with NAFLD who subsequently underwent cardiorespiratory polygraphy to detect OSA, the prevalence of significant thickening of CIMT was higher in patients at high risk compared to those at low risk for OSA (51.2% vs 24.3%, p = 0.006).

"Furthermore, our data demonstrated a significant rise in ICAM-1 levels with severity of OSA as categorized by

increasing AHI, from mild 259.16 ± 57.40 ng/mL, moderate 299.65 ± 90.41 ng/mL and severe 370.63 ± 57.71 ng/mL, p < 0.001". 

This data has not been presented anywhere in the results section.

Thank you for the comment. You are absolutely right, we have not presented this data in the result section. We have removed this statement in the discussion section.

Reference no 35 has been removed. Subsequent references have been adjusted accordingly.

The discussion is too long with many repetitions and can be condensed.

Thank you for the comment. The discussion has been refined to avoid repetitions without losing the relevant findings of the study.

 

Reviewer #2: This was an interesting study that examined potential links between OSA, liver disease and CV risks. The authors studied 110 patients with suspected OSA. Authors found that OSA severity was significantly associated with liver disease. CV risk (CIMT) was elevated in liver patients. I had the following questions/comments:

1. In the discussion, other limitations that should be mentioned include: relatively small numbers, single centre, cross sectional nature

Thank you for the comment. This has been added in the limitation. . “The single center data collection, the cross-sectional design and the relatively small number of subjects are areas of improvement for future studies.”

2. I wasn’t sure if these were consecutive patients? if not, how were they selected?

Thank you for the question. Patients were recruited based on convenient sampling. This has been added in the method section.” We screened 110 subjects consecutively, between the ages of 18 to 65, who were diagnosed with OSA….

3. When was the blood taken in relation to PSG (i.e. time frame)?

Thank you for the question. PSG was done at a much earlier date just to establish the diagnosis of OSA. 

4. I assume none of these patients were treated?

Thank you for the question. Upon recruitment, the patients have not initiated treatment yet. 

5. Was there an analysis of desaturation indices (e.g. 4% ODI, mean saturation, % time below 90% etc). I would suspect these might be highly related to liver disease?

Thank you for the question. We understand the relevance. Unfortunately, these were not available in our analysis.

6. I wasn’t sure why ICAM specifically was the inflammatory marker selected?

Thank you for the question. We acknowledge the many surrogate markers for cardiovascular disease including ICAM-1, which is an intercellular adhesion molecule that are expressed on the surface of vascular endothelial cells. It is a valid and recognised research tool, serving as one of the many molecular markers for atherosclerosis and the development of coronary heart disease.

7. Mean BMI, AHI, saturation, etc should be added to Table 1. Did any patients have obesity hypoventilation syndrome? This might have increased right sided pressures and lead to more liver swelling?

Thank you for the suggestion. Mean BMI and AHI has been added in Table 1. But we were unable to include the saturation indices. The study also did not look into obesity hypoventilation syndrome and its associated complications. This would however be an interesting variable for future studies in the area.

8. There should be more description of the ultrasound methods to detect liver disease. How validated are they in the context of substantial obesity +/- high right sided pressures?

Thank you for the suggestion. We have added the this statement. “Evidence of NAFLD was confirmed with radiological technique of liver-kidney contrast on degree of stetatosis and further divided into four grades, severe (NAFLD-3), moderate (NAFLD-2), mild (NAFLD-1) and normal (non-NAFLD)…..” on Page 9. The examinations were performed by 2 independent, trained radiologists based according to universally accepted criteria (Joy, D., Thava, V. R., & Scott, B. B. (2003). Diagnosis of fatty liver disease: is biopsy necessary? Eur J Gastroenterol Hepatol, 15(5), 539-543. doi:10.1097/01.meg.0000059112.41030.2e)

9. In the last paragraph of page 8, do you mean carotid ultrasound or abdominal?

Thank you for highlighting this. We have added – “For the CIMT measurements subjects were also scanned in the supine position by two independent radiologists utilizing the same equipment. The distance between the 2 lines gave a reliable index of the thickness of the carotid intimal-medial complex. All measurements were made at the time of the scan on frozen images of longitudinal scans by using the machine’s electronic caliper. Carotid segments for far (posterior) walls of each common carotid artery at a distance of 1cm from the bulb will be examined. The average of right and left CIMT were calculated and were recorded into millimeters (mm)….”

10. Was there an analysis of BMI, AHI as continuous variables? It was difficult to understand if Tables 7 and 8 reflected multivariate models? This should be made clearer.

Thank you for the comment. There were analysis of BMI and AHI as continuous variables and we have added the mean±SD in Table 1. We have removed Tables 5 and 7 as suggested by Reviewer 1. Table 8 has been adjusted to Table 6. It is a multivariate analysis. 

11. The CI around the OR for severe OSA is very wide-- these should be added to the abstract so readers understand the instability of these measures.

Thank you for the suggestion. This has been added in the abstract. 

12. Was there a multivariate analysis looking at determinants of CIMT? I think you need this to determine whether liver disease is an independent risk factor.

Thank you for the suggestion. This was not assessed in this paper but worthwhile addressing in future studies.

Thank you again for your suggestions. We look forward to your positive review.

---

## [Decision Letter · Decision Letter 1]

3 May 2021

PONE-D-21-03573R1

Significantly Higher Atherosclerosis Risks in Patients with Obstructive Sleep Apnoea and Non Alcoholic Fatty Liver Disease

PLOS ONE

Dear Dr. Abdul Ghani,

Thank you for submitting your manuscript to PLOS ONE. After careful consideration, we feel that it has merit but does not fully meet PLOS ONE’s publication criteria as it currently stands. Therefore, we invite you to submit a revised version of the manuscript that addresses the points raised during the review process.

Please note that Reviewer #2 has two additional comments to be addressed. I concur with both. 

We look forward to receiving your revised manuscript.

Kind regards,

James Andrew Rowley

Academic Editor

PLOS ONE

Journal Requirements:

Reviewers' comments:

Reviewer's Responses to Questions

**Comments to the Author**

1. If the authors have adequately addressed your comments raised in a previous round of review and you feel that this manuscript is now acceptable for publication, you may indicate that here to bypass the “Comments to the Author” section, enter your conflict of interest statement in the “Confidential to Editor” section, and submit your "Accept" recommendation.

Reviewer #1: All comments have been addressed

Reviewer #2: (No Response)

2. Is the manuscript technically sound, and do the data support the conclusions?

Reviewer #1: Yes

Reviewer #2: Partly

3. Has the statistical analysis been performed appropriately and rigorously? 

Reviewer #1: Yes

Reviewer #2: I Don't Know

4. Have the authors made all data underlying the findings in their manuscript fully available?

Reviewer #1: Yes

Reviewer #2: No

5. Is the manuscript presented in an intelligible fashion and written in standard English?

Reviewer #1: Yes

Reviewer #2: No

6. Review Comments to the Author

Reviewer #1: (No Response)

Reviewer #2: The authors have addressed the majority of my comments and the paper improved. I just had afew more:

1. I wasnt sure why desaturation parameters were not available? Isnt this part of the PSG report (event indices such as mean saturation, % below 90% would have been useful).

2. It seems like the blood samples were taken after PSG. To clarify, were any of the patients on treatment at the time of the blood samples (it was unclear to me when recruitment was)?

7. PLOS authors have the option to publish the peer review history of their article (what does this mean?). If published, this will include your full peer review and any attached files.

Reviewer #1: **Yes: **Swastik Agrawal

Reviewer #2: No

---

## [Author Response · Author response to Decision Letter 1]

4 May 2021

Dear Sir, 

Thank you again for your kind responses and effort towards this manuscript. I hereby provide the details of the amendments as required by the Editors and Reviewer 2. 

4. Have the authors made all data underlying the findings in their manuscript fully available?

Reviewer #1: Yes

Reviewer #2: No

Author response: The raw data has been uploaded as supporting document.

5. Is the manuscript presented in an intelligible fashion and written in standard English?

Reviewer #1: Yes

Reviewer #2: No

Author response: The manuscript has been checked, corrected and revised thoroughly. 

Reviewer 2:

The authors have addressed the majority of my comments and the paper improved. I just had a few more:

1. I wasn’t sure why desaturation parameters were not available? Isn’t this part of the PSG report (event indices such as mean saturation, % below 90% would have been useful).

Thank you for the question. I apologize for the ambiguous response previously. 

The data is available. However, in relation to the current data presented in this manuscript, the PSG served only as a diagnostic tool to identify patients with OSA and subsequent categorization on severity based on the AHI. Further discussion on OSA would make the manuscript too lengthy and subsequently exceed the stipulated maximum word count.

However, we do acknowledge the relevance of the event indices that you have mentioned above and we enthuse over a future report to determine the associations between those indices and the studied atherosclerosis risks.

2. It seems like the blood samples were taken after PSG. To clarify, were any of the patients on treatment at the time of the blood samples (it was unclear to me when recruitment was)?

Thank you for the question. All patients were recruited prior to initiation of treatment (if deemed necessary by the treating physician). 

This statement has been added to the methodology section. “Patients who fulfilled the inclusion criteria were approached, explained and informed regarding the main objectives of the study prior to the initiation of mechanical ventilation, if required.” 

Thank you again for your suggestions. We look forward to your positive review.

---

## [Decision Letter · Decision Letter 2]

13 May 2021

PONE-D-21-03573R2

Significantly Higher Atherosclerosis Risks in Patients with Obstructive Sleep Apnoea and Non Alcoholic Fatty Liver Disease

PLOS ONE

Dear Dr. Abdul Ghani,

Thank you for submitting your manuscript to PLOS ONE. After careful consideration, we feel that it has merit but does not fully meet PLOS ONE’s publication criteria as it currently stands. Therefore, we invite you to submit a revised version of the manuscript that addresses the points raised during the review process.

Please note that while Reviewer #2 is most satisfied with your response, they still have two points to clarify.  Please do so in your response and in the manuscript. 

We look forward to receiving your revised manuscript.

Kind regards,

James Andrew Rowley

Academic Editor

PLOS ONE

Journal Requirements:

Reviewers' comments:

Reviewer's Responses to Questions

**Comments to the Author**

1. If the authors have adequately addressed your comments raised in a previous round of review and you feel that this manuscript is now acceptable for publication, you may indicate that here to bypass the “Comments to the Author” section, enter your conflict of interest statement in the “Confidential to Editor” section, and submit your "Accept" recommendation.

Reviewer #2: (No Response)

2. Is the manuscript technically sound, and do the data support the conclusions?

Reviewer #2: Yes

3. Has the statistical analysis been performed appropriately and rigorously? 

Reviewer #2: Yes

4. Have the authors made all data underlying the findings in their manuscript fully available?

Reviewer #2: (No Response)

5. Is the manuscript presented in an intelligible fashion and written in standard English?

Reviewer #2: Yes

6. Review Comments to the Author

Reviewer #2: Thank you for the responses- a minor point. "Patients who fulfilled

the inclusion criteria were approached, explained and informed regarding the main objectives

of the study prior to the initiation of mechanical ventilation, if required." Can you change mechanical ventilation to "positive airway pressure". Can you also explicitly state that samples were taken prior to initiation of therapy (this only says they were recruited prior to treatment).

7. PLOS authors have the option to publish the peer review history of their article (what does this mean?). If published, this will include your full peer review and any attached files.

Reviewer #2: No

---

## [Author Response · Author response to Decision Letter 2]

17 May 2021

Dear Sir, 

Thank you again for your kind responses and effort towards this manuscript. I hereby provide the details of the amendments as required by Reviewer 2. 

Reviewer #2: Thank you for the responses- a minor point. "Patients who fulfilled

the inclusion criteria were approached, explained and informed regarding the main objectives

of the study prior to the initiation of mechanical ventilation, if required." Can you change mechanical ventilation to "positive airway pressure". Can you also explicitly state that samples were taken prior to initiation of therapy (this only says they were recruited prior to treatment).

This sentence has been further amended to: “Patients who fulfilled the inclusion criteria were approached, explained and informed regarding the main objectives of the study and blood samples were subsequently taken, prior to the initiation of positive airway pressure, if required.”

Thank you again for your suggestions. We look forward to your positive review.

---

## [Editor Report · Decision Letter 3]

2 Jun 2021

Significantly Higher Atherosclerosis Risks in Patients with Obstructive Sleep Apnea and Non Alcoholic Fatty Liver Disease

PONE-D-21-03573R3

Dear Dr. Abdul Ghani,

We’re pleased to inform you that your manuscript has been judged scientifically suitable for publication and will be formally accepted for publication once it meets all outstanding technical requirements.

Kind regards,

James Andrew Rowley

Academic Editor

PLOS ONE
---

## [Editor Report · Acceptance letter]

22 Jun 2021

PONE-D-21-03573R3 

Significantly Higher Atherosclerosis Risks in Patients with Obstructive Sleep Apnea and Non-Alcoholic Fatty Liver Disease 

Dear Dr. Abdul Ghani:

I'm pleased to inform you that your manuscript has been deemed suitable for publication in PLOS ONE. Congratulations! Your manuscript is now with our production department. 

Kind regards, 

on behalf of

Dr. James Andrew Rowley 

Academic Editor

PLOS ONE